# Characterization of Cellular and Humoral Immunity to Commercial Cattle BVDV Vaccines in White-Tailed Deer

**DOI:** 10.3390/vaccines13040427

**Published:** 2025-04-18

**Authors:** Paola M. Boggiatto, Mitchell V. Palmer, Steven C. Olsen, Shollie M. Falkenberg

**Affiliations:** 1Infectious Bacterial Diseases Research Unit, National Animal Disease Center, 1920 Dayton Avenue, Ames, IA 50010, USA; mitchell.palmer@usda.gov (M.V.P.);; 2Department of Pathobiology, Auburn University College of Veterinary Medicine, 1130 Wire Road, Auburn, AL 36849, USA

**Keywords:** white-tailed deer, bovine viral diarrhea virus, BVDV, immune responses, PrimeFlow, vaccines

## Abstract

Background/Objectives: White-tailed deer (*Odocoileus virginianus*) (WTD) play a central role at the human–livestock–wildlife interface, given their contribution to the spread of diseases that can affect livestock. These include a variety of bacterial, viral, and prion diseases with significant economic impact. Given the implications for WTD as potential reservoirs for a variety of diseases, methods for prevention and disease control in WTD are an important consideration. Methods: Using commercial livestock vaccines against bovine viral diarrhea virus (BVDV) in killed and modified live formulations, we test the ability of WTD to develop humoral and cellular immune responses following vaccination. Results: We demonstrate that, similar to cattle, WTD develop humoral immune responses to both killed and modified live formulations. Conclusions: As the farmed deer industry and the use of livestock vaccines in non-approved species grow, this type of information will help inform and develop improved husbandry and veterinary care practices. Additionally, while we were unable to detect cell-mediated immune responses to the vaccine, we established PrimeFlow as a method to detect IFN-γ responses in specific T cell populations, adding another level of resolution to our ability to understand WTD immune responses.

## 1. Introduction

White-tailed deer (*Odocoileus virginianus*) (WTD) are the most abundant wild ungulate in North America. The transmission of SARS-CoV-2, the COVID-19 virus, between humans and deer [1], as well as the transmission of tuberculosis between deer and cattle [2], demonstrate the central role WTD play at the human–livestock–wildlife interface. Furthermore, farmed WTD may play a dynamic role in zoonotic pathogen transmission, as deer are susceptible to Q-fever, leptospirosis, salmonellosis, cryptosporidiosis, and giardiasis [3].

Chronic wasting disease (CWD), a spongiform encephalopathy, is present in both free-ranging and farmed deer but has not been documented in humans or livestock. Since its first report in 1967, CWD has spread to 26 US states and three Canadian provinces [4]. Additionally, captive wild ruminants in the United States including sambar deer (*Cervus unicolor*), WTD, waterbuck (*Kobus ellipsiprymnus*), and elk (*Cervus elaphus*), harbor coronaviruses (CoVs) that are biologically, genetically, and antigenically closely related to bovine CoV (BCoV) [5]. Lastly, WTD have been reported to be susceptible to bovine viral diarrhea virus (BVDV) [6,7] and can produce persistently infected (PI) fawns [8], thus serving as a source for BVDV infection. BVDV, a pestivirus belonging to the *Flaviviridae* family, is a viral pathogen of cattle resulting in significant losses to the industry, primarily through reproductive failure. This economically important pathogen has been documented to be spread between WTD and cattle [9]. Vaccination is a critical part of controlling BVDV and decreasing the frequency of reproductive losses due to BVDV infections. Vaccination in cattle has been shown to result in protective responses associated with virus-neutralizing antibody titers [10,11] as well as cell-mediated immune (CMI) responses, characterized by interferon gamma (IFN-γ)-mediated T helper 1 responses [12,13,14].

Given the susceptibility of WTD to a variety of diseases, methods for prevention and disease control in WTD are an important consideration. Currently in the United States, there are approximately 4000 white-tailed deer agricultural operations with an approximate revenue of USD 44 million annually [15]. Knowledge of deer health is imperative to maintain such operations, and routine vaccinations are part of any herd health program. Typically, ruminant vaccines used in WTD are used off-label, as no preventive label claims are specific for WTD.

In general, knowledge of vaccine-induced immune responses or vaccine efficacy is lacking for most wildlife species. Given the use of commercial vaccines in farmed WTD and their susceptibility and potential role in transmission for various diseases, understanding if and how WTD respond to these vaccines is critical for our ability to enhance animal health and implement disease management strategies. To bridge this knowledge gap, we use two licensed vaccines against BVDV commonly used in cattle and assess immune responses in WTD. Commercial vaccines used in the United States are multivalent, containing both viral and bacterial targets, and come either as modified live vaccines (MLV) or killed viral vaccines (KV) formulations. In this study, we assess both humoral and cell-mediated (CMI) immune responses in WTD following vaccination. Specifically, we focused on BVD virus neutralization titers and BVDV-specific IFN-γ T cell responses as shown previously [16].

## 2. Materials and Methods

### 2.1. Animals, Vaccination, and Sample Collection

White-tailed deer were hand-raised and housed at the National Animal Disease Center (NADC) in Ames, Iowa. All work involving animals was conducted in accordance with recommendations in the Care and Use of Laboratory Animals of the National Institutes of Health and the Guide for the Care and Use of Agricultural Animals in Research and Teaching. Additionally, all animal-related procedures were approved by and carried out under the NADC Institutional Animal Care and Use Committee (IACUC) guidelines and oversight. Additionally, all procedures involving animals were approved by the NADC IACUC (Protocol number: ARS-2020-909). Eighteen 7–9-month-old female and castrated male white-tailed deer were divided into 3 experimental groups: non-vaccinated controls (n = 6), killed vaccine (KV) (n = 6), and modified live vaccine (MLV) (n = 6). Deer in the KV group were vaccinated with the commercially available multivalent vaccine Vira Shield™ 6 (Elanco Animal Health, Inc., Greenfield, IN, USA) containing BVDV type 1 and type 2, bovine herpesvirus-1 (BHV-1), bovine respiratory syncytial virus (BRSV), and parainfluenza-3 (PI-3). Deer in the MLV group were vaccinated with the commercially available pentavalent vaccine Bovi-Shield Gold 5 (Zoetis, Inc., Parsippany, NJ, USA) containing BVDV type 1 and type 2, BHV-1, BRSV, and PI-3. The two vaccines were administered at the recommended volume and per the recommended route, in the cervical area, according to the manufacturers’ instructions. Approximately 4 weeks post-initial vaccination, deer in the KV group were re-vaccinated, again, per label dose and route of administration, according to the manufacturer’s instructions. Blood samples were collected from the jugular vein prior to initial vaccination and at approximately 4 and 8 weeks post-initial vaccination.

### 2.2. BVDV Strains

For in vitro stimulation of peripheral blood mononuclear cells (PBMCs), field strains of non-cytopathic BVDV-1a (PI34) and BVDV-2a (PI28) were utilized as previously described for cattle [17]. For assessment of serological responses, cytopathic reference strains BVDV-1a (Singer) and BVDV-2a (296c) were used. Propagation of strains was performed as previously described [18]. Viral titers were determined through serial 10-fold dilutions, individually run in quintuplets on bovine turbinate (BTu) cells. Cytopathic effect (CPE) was evaluated as previously described [19].

### 2.3. Serology

To assess responses to vaccination, titers to bovine viral diarrhea virus 1 and 2 (BVDV-1 and BVDV-2) were measured from serum samples. Blood was collected into serum separation tubes and centrifuged at 1200× *g* at room temperature for 25 min. Serum was then aliquoted and stored at −80 °Celsius (C) until analysis. Evaluation of titer responses to vaccination was performed as assessed by determination of a virus neutralization titer (VNT) using cytopathic isolates BVDV-1a-Singer and BVDV-2a-296c as described [18]. Briefly, serum samples were serially diluted from 1:2 to 1:4096 in minimal essential media (MEM) using flat-bottom 96-well plates, and 200 TCID_50_ of virus was added to each well. Samples and virus were incubated for 1 h at 37 °C with 5% CO_2_. Following incubation, 2 × 10^5^ BTu cells were added to each well and incubated for 4 days at 37 °C with 5% CO_2_. Dilution endpoints were determined by observation of cytopathic effect (CPE) in the BTu cell monolayer. Results were expressed as the antilog base 10 of the highest serum dilution able to inhibit CPE in the monolayer. Sample dilutions were run in triplicates and VNT was calculated by the Spearman–Karber method.

### 2.4. Peripheral Blood Mononuclear Cell (PBMC) Isolation and Stimulation

Blood samples were collected into acid citrate dextrose (ACD) tubes, and PBMCs were isolated using SepMate tubes according to the manufacturer’s recommendations (Stemcell Technologies, Cambridge, MA, USA), as described previously [16,17]. Following isolation, PMBCs were passed through a 40 μm filter to remove cellular debris. Viability and counts were then determined via the Muse™ Cell analyzer (Cytek, Fremont, CA, USA) according to the manufacturer’s recommendations. PMBCs were adjusted to 1 × 10^7^ cells per ml, and approximately 1 × 10^6^ live cells in 100 μL of cell suspension were added onto 96-well round-bottom plates containing 100 μL of complete RPMI medium, supplemented as described previously [16]. All samples were plated in duplicate for each stimulation condition, and plates were incubated at 37 °C with 5% CO_2_.

### 2.5. In Vitro Stimulation Conditions

Mitogen and antigen stimulations were performed as previously described [20]. Briefly, as a positive control for stimulation, 50 μL of media was removed from the respective wells and replaced with 50 μL of cell stimulation cocktail (Phorbol myristate acetate (PMA) and ionomycin; 8 μL diluted in 1 mL of complete RPMI media; (eBioscience, San Diego, CA, USA). For antigen stimulations, 50 μL of media was removed from the respective wells and replaced with 50 μL of BVDV-1a (PI34) or 50 μL of BVDV-2a (PI28), at a multiplicity of infection (MOI) of approximately 1. Non-stimulated wells were left untouched. Following the addition of mitogen or antigen, plates were returned to the incubator and maintained at 37 °C with 5% CO_2_. Stimulation times were approximately 1.5 h for mitogen and approximately 24 h for antigen.

### 2.6. Staining for Flow Cytometry and PrimeFlow

Cells were processed for flow cytometry analysis as previously described [20]. Briefly, plates were removed from the incubator and centrifuged at 300× *g* for 4 min at room temperature. The supernatant was discarded, and all PMBCs were resuspended in 100 μL of phosphate-buffered saline (PBS). At this time duplicate wells were combined, and the pooled samples transferred to a new 96-well round-bottom plate for further processing. PBMCs were then washed twice at 300× *g* for 4 min and then incubated with 50 μL of a 1:100 dilution of a primary monoclonal antibody cocktail against CD4 (goat, Clone 17D, IgG1), CD8 (sheep, Clone ST8, IgM), and γδ (bovine, Clone GB21A, IgG2b) for 15 min at room temperature (RT) (Washington State Monoclonal Antibody Center, Pullman, WA, USA). After the incubation period, 50 μL of a cocktail including the following fluorescent-labeled secondary antibodies were added: anti-IgG1 BUV395 (BD Bioscience, San Diego, CA, USA), FITC-labeled anti-IgM (Biolegend, San Diego, CA, USA), and BV711-labeled anti-IgG2a (BD Bioscience). Secondary antibodies were incubated for 15 min at RT in the dark. PMBC were then washed twice in PBS as described above and further processed for the PrimeFlow assay for IFN-γ detection as previously described [16,17]. Proprietary specific oligonucleotide (RNA) probes designed specially to detect WTD IFN-γ (6007031-210V) labeled with AF750 were designed by Thermo Fisher Scientific. All data was collected using the BD FACS Symphony flow cytometer (BD Biosciences, San Diego, CA, USA).

### 2.7. Data Analysis

The frequency of each T cell subset (CD4, CD8, or γδ) and the frequencies of IFN-γ mRNA-expressing T cell populations were determined using FlowJo™ software (FlowJo, LLC, BD Bioscience, Ashland, OR, USA). Average VNT was determined for each respective group using individual VNT values for each respective animal at each respective timepoint. As previously described in VNT methods, the neutralization results of the 3 wells at each respective dilution were evaluated for neutralization or lack of CPE. These results were calculated for the VNT and were used to convert to a log_2_ value according to the Spearman–Karber method. All data were graphed and analyzed using GraphPad Prism 9 (GraphPad Software, Boston, MA, USA). Statistical significance was evaluated using a 2-way ANOVA with multiple comparisons with a Tukey’s correction.

## 3. Results

### 3.1. Neutralizing Antibody Titers to BVDV-1 and BVDV-2a

We assessed serological responses to BVDV-1a and BVDV-2a via VNT. Shown in Figure 1 are individual and average VNT values for each experimental group. Prior to vaccination, none of the deer had any serological responses to either strain tested. At 4 weeks post-vaccination, we observed measurable VNT to BVDV-1 following MLV vaccination, and by 8 weeks post-vaccination, titers to BVDV-1 were observed in both KV and MLV vaccination groups (Figure 1A). Titers to BVDV-2a were only observed in the MLV vaccination group, at both 4 and 8 weeks post-vaccination (Figure 1B). These findings are consistent with the BVDV strains present in each vaccine used in this study, with the KV containing BVDV-1 and the MLV containing BVDV-2a. Overall, MLV-vaccinated animals demonstrated a greater serological response at the last time point tested, with a mean VNT of 51.9 ± 0.467 to BVDV-2a, as compared to the KV-vaccinated animals, with a mean VNT of 32 ± 10.9 to BVDV-1. Additionally, statistically significant differences in BVDV-2a VNT were only observed for the MLV-vaccinated animals as compared to both control and KV groups (Figure 1B).

### 3.2. Frequency of Circulating T Cell Subsets and Antigen-Specific IFN-γ Responses

Peripheral blood mononuclear cells (PMBCs) were isolated from whole blood samples collected at 8 weeks post-vaccination, stimulated in vitro, and analyzed via flow cytometry for frequencies of CD4, CD8, and γδ T cell subsets (Figure 2A, representative dot plots). Overall, CD4^+^ T cells constituted the majority of circulating PBMCs (~60%), followed by CD8^+^ T cells (~8%) and γδ T cells (~10%) (Figure 2B). Frequencies of these different T cell subsets were similar across experimental groups (Control, KV, and MLV), suggesting that vaccination did not significantly affect the distribution of these cells, at least at the time point analyzed.

To assess cell-mediated responses, particularly IFN-γ responses, we utilized PrimeFlow followed by flow cytometry to measure intracellular IFN-γ mRNA. As previously described [16], PrimeFlow utilizes probes to detect target mRNA, which can be designed to be species-specific. Therefore, we designed probes to detect WTD IFN-γ mRNA. Following PMA/ionomycin stimulation as a positive control, we observed an increase in the frequency of IFN-γ mRNA-expressing PMBC as compared to non-stimulated PBMCs (Figure 3A, representative dot plots). Additionally, when combined with surface staining for CD4 (Figure 3B), CD8 (Figure 3C), and γδ (Figure 3D), we observe that all three T cell subsets are capable of expressing IFN-γ mRNA in response to stimulation. Altogether, these data demonstrate that this approach can successfully detect IFN-γ mRNA in white-tailed deer PMBC.

We then applied this approach to assess antigen-specific IFN-γ responses following vaccination. Following in vitro stimulation with BVDV-1 or BVDV-2a, we did not observe any significant changes in the frequency (Figure 4A) or number (Figure 4B) of CD4^+^ T cells expressing IFN-γ mRNA in vaccinated animals as compared to control animals. Similarly, we did not observe any significant changes in the frequency or number of CD8^+^ (Figure 4C,D) or γδ (Figure 4E,F) T cells expressing IFN-γ mRNA in response to BVDV antigen stimulation. Altogether, these data would suggest that while this assay can detect IFN-γ mRNA, neither KV nor MLV vaccinations induced an increased IFN-γ response at the time point analyzed.

## 4. Discussion

In this study we set out to assess the serological and cellular immune responses induced by vaccination with commercial BVDV vaccines for domestic livestock in WTD. The data presented demonstrates that KV induces measurable antibody titers to BVDV-1, while MLV can induce measurable antibody titers against BVDV-1 and BVDV-2, albeit with higher titers against BVDV-2. Similar to antibody responses to BVDV in cattle following vaccination, antibody titers to the MLV appear earlier, at approximately 4 weeks post-vaccination, while measurable antibody titers following inoculation with the KV do not appear until 8 weeks post-vaccination, 4 weeks following the manufacturer’s recommended boost. These findings, along with the skewed antigen differences, are consistent with previously reported data from cattle vaccinated with these same vaccines [20], suggesting similar immunogenicity in inducing humoral responses in cattle and WTD.

Assessment of CMI responses can sometimes be hindered by the lack of reagents available for use in non-traditional laboratory species. The assay used in this study to measure cellular responses following vaccination utilizes oligonucleotide probes to detect target gene products. While this requires knowledge of the sequence for the gene of interest, generating probes is faster and easier than generating antibodies against a specific protein target. We combined traditional cellular staining with commercially available antibodies against T cell surface markers with intracellular detection of host IFN-γ mRNA using oligonucleotide probes, PrimeFlow. When we stimulated WTD PBMCs with a mitogen to induce IFN-γ production, our assay detected IFN-γ mRNA from various T cell subsets, including CD4 and CD8 T cells. To our knowledge, this is the first time that PrimeFlow was utilized to detect intracellular IFN-γ mRNA using WTD PBMC. Characterization of IFN-γ production in the context of CMI is important, as this cytokine is a key player in the orchestration of anti-viral, anti-bacterial, and anti-parasitic infections [21]. Since multiple cell types can produce IFN-γ, ELISAs do not provide information as to the cellular source of this cytokine. Therefore, development of an assay, such as the one described here, that can determine the source of IFN-γ provides additional information regarding the nature of the cellular immune response.

Unlike the response observed with mitogen stimulation, we were not able to detect BVDV-specific IFN-γ mRNA in any of the T cell populations analyzed following vaccination. The time point chosen for analysis was based on previous experience with analysis of T cell responses in cattle. It is possible that this was not an appropriate timepoint for measuring T cell responses in WTD and that a later timepoint (i.e., 12 or 16 weeks post-vaccination) would have allowed detection. One of the major limitations of this study was the limited time points used to assess CMI. Therefore, we cannot conclusively state that CMI responses to the vaccines were not elicited. However, this limitation can be overcome in subsequent studies analyzing different timepoints and/or by optimizing antigen stimulation conditions.

## 5. Conclusions

Altogether, the data presented here demonstrate that vaccination with commercially available livestock vaccines can be effective at inducing measurable humoral responses in WTD. However, it should be noted, especially in the case of the MLV, that additional studies to assess the safety of a live vaccine in a new species should be carried out. Furthermore, both vaccines used in this study are multivalent vaccines for use against other pathogens, including bovine herpesvirus-1 (BHV-1), bovine respiratory syncytial virus (BRSV), and parainfluenza-3 (PI-3). Our work was limited to analyzing responses to BVDV, but knowledge of WTD response to these additional pathogens would further enhance our understanding of how these vaccines shape the WTD immune response. As the farmed deer industry continues to grow, along with the use of livestock vaccines in non-approved species, this information is valuable to producers and veterinarians alike as husbandry and veterinary care practices are developed and modified. Here we established PrimeFlow as a method to detect IFN-γ responses in specific T cell populations, adding another level of resolution to our ability to understand WTD immune responses.

## Figures and Tables

**Figure 1 vaccines-13-00427-f001:**
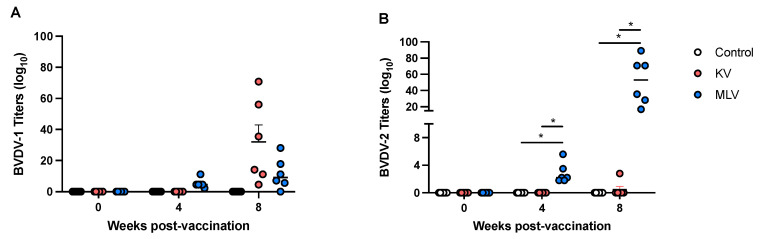
Serological responses to KV and MLV vaccines in serum of vaccinated white-tailed deer. Serum samples were collected at the indicated timepoints following vaccination and boost and assessed via virus neutralization assay. Neutralizing titers were determined for BVDV-1 (**A**) and BVDV-2 (**B**). Titers are expressed as the reciprocal (anti-log base 10) of the highest serum dilution able to inhibit CPE in the monolayer. * Indicates statistical significance (*p* ≤ 0.05).

**Figure 2 vaccines-13-00427-f002:**
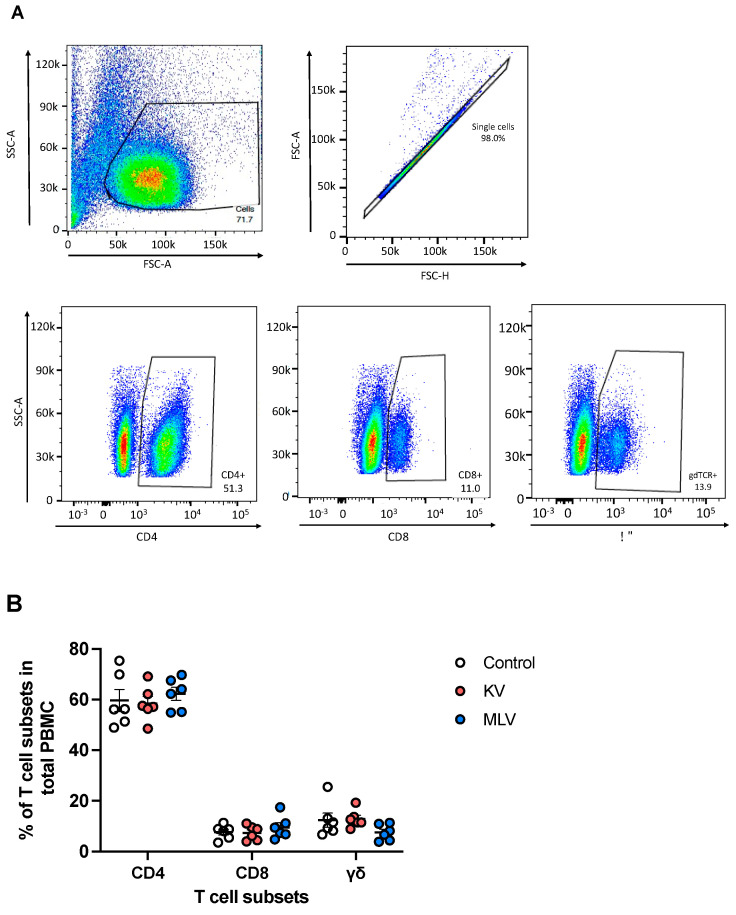
Distribution of peripheral T cell subsets following vaccination of white-tailed deer with commercial vaccines. Peripheral blood mononuclear cells (PBMC) were isolated from whole blood samples and assessed via flow cytometry for the frequency of CD4, CD8, and γσ T cells using flow cytometry. Shown are representative dot plots (**A**) for forward scatter (FSC) vs. side scatter (SSC), singlet discrimination, CD4, CD8, and γδ gating. Overall frequency of T cell subsets control (open circle), KV-vaccinated deer (orange), and MLV-vaccinated deer (blue) (**B**). Values indicate mean frequency ± SEM.

**Figure 3 vaccines-13-00427-f003:**
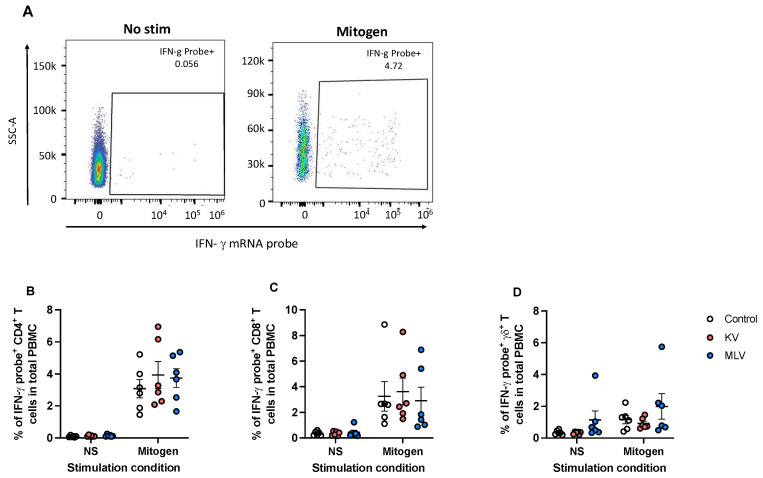
Detection of intracellular IFN-γ mRNA in white-tailed deer T cells using PrimeFlow. Peripheral blood mononuclear cells were isolated from whole blood samples and were either left unstimulated or stimulated with PMA/Ionomycin to induce cytokine production. PrimeFlow and flow cytometry were utilized to assess IFN-γ mRNA expression from CD4, CD8, and γδ T cells. Shown are the representative dot plots (**A**) of total PBMCs with and without PMA/Ionomycin stimulation, demonstrating the presence of IFN-γ mRNA staining. The frequencies of IFN-γ mRNA-positive CD4 (**B**), CD8 (**C**), and γδ (**D**) T cells from PBMCs collected from control (open circles), KV-vaccinated deer (orange circles), and MLV-vaccinated deer (blue circles) and stimulated with PMA/Ionomycin are shown. Values indicate mean frequency ± SEM.

**Figure 4 vaccines-13-00427-f004:**
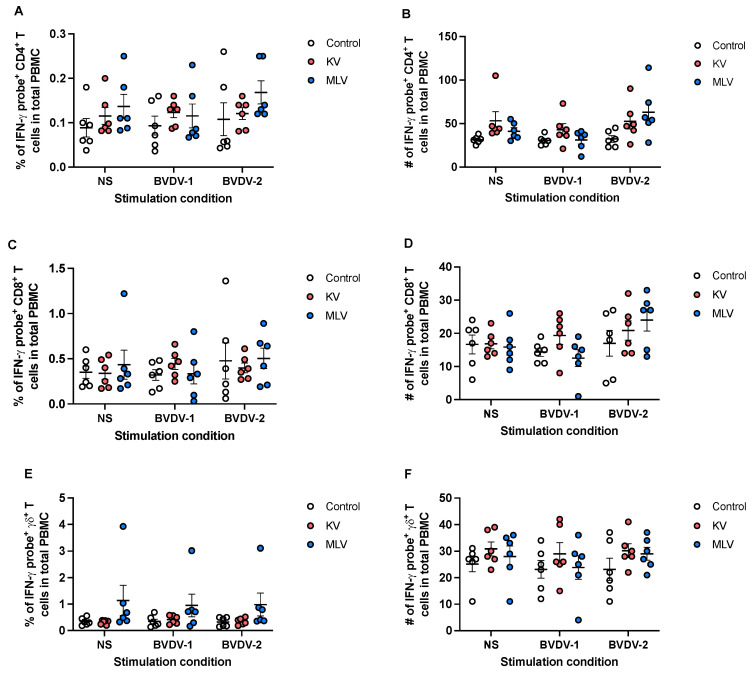
Assessment of BVDV-specific IFN-γ T cell responses from vaccinated white-tailed deer. Peripheral blood mononuclear cells were isolated from whole blood at 8 weeks post-vaccination and were left unstimulated or were stimulated with BVDV-1 or BVDV-2a in vitro. Using PrimeFlow and flow cytometry, the frequency and number of IFN-γ mRNA-expressing CD4 (**A**,**B**), CD8 (**C**,**D**), and γδ (**E**,**F**) T cells from control (open circles), KV-vaccinated (orange circles), and MLV-vaccinated (blue circles) were assessed. Values indicated mean frequency ± SEM.

## Data Availability

All data generated are presented in the manuscript.

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
