# Peer review of "Characterization of Cellular and Humoral Immunity to Commercial Cattle BVDV Vaccines in White-Tailed Deer"

_vaccines, 2025, doi:10.3390/vaccines13040427_

Round 1
Reviewer 1 Report
Comments and Suggestions for Authors
The manuscript, “Characterization of cellular and humoral immunity to commercial cattle BVDV vaccines in white-tailed deer,” is a very interesting study that provides important information for cattle and deer farms.
The manuscript is well-written and easy to understand. The first two paragraphs of the introduction explain different diseases that might be caused by deer and their consequences for other species. As a suggestion, more emphasis could be placed on BVDV, but this should not compromise the manuscript.
My main concerns are: Approving a live vaccine in a different species caught my attention. In my understanding, only killed antigens should be used. The second is the antibodies used in flow-cytometry; those are specifically for deer or have been found to cross-react with deer molecules.
The manuscript can be published.
Reviewer 2 Report
Comments and Suggestions for Authors
Boggiatto et al. tested the ability of white-tailed deer to develop humoral and cellular responses following immunization with two commercially available livestock vaccines.
Although the main message of the manuscript is straightforward, the authors could have provided more context and discussion so that readers from a broader background could benefit from the paper.
Specific comments:
- In the discussion, the authors wrote, “the data presented demonstrates that KV and MLV can induce measurable antibody titers against BVDV1 and BVDV2, respectively,” while KV does not induce measurable antibody titers against BVDV2, according to Figure 1b.
- Given that KV and MLV induced different humoral immune responses against BVDV1 and BVDV2, the humoral responses against other targets in the vaccines [bovine herpes virus-1 (BHV-1), bovine respiratory syncytial virus (BRSV), and parainfluenza-3 (PI-3)] are worth exploring or at least considering in the discussion.
- Given that KV and MLV induced different humoral immune responses against BVDV1 and BVDV2, the difference between KV and MLV warrants discussion (e.g., is the modified live vaccine expected to infect WTD?).
- It is unclear why T-cell responses are measured and what type of T-cell responses are expected based on experience in cattle.
